# Magnolol Supplementation Alters Serum Parameters, Immune Homeostasis, Amino Acid Profiles, and Gene Expression of Amino Acid Transporters in Growing Pigs

**DOI:** 10.3390/ijms241813952

**Published:** 2023-09-11

**Authors:** Yanchen Liu, Yuanfei Li, Miao Yu, Zhimei Tian, Jinping Deng, Xianyong Ma, Yulong Yin

**Affiliations:** 1Guangdong Provincial Key Laboratory of Animal Nutrition Control, National Engineering Research Center for Breeding Swine Industry, Institute of Subtropical Animal Nutrition and Feed, College of Animal Science, South China Agricultural University, Guangzhou 510642, China; mryanchenliu@outlook.com (Y.L.); dengjinping@scau.edu.cn (J.D.); 2State Key Laboratory of Swine and Poultry Breeding Industry, Institute of Animal Science, Guangdong Academy of Agricultural Sciences, Guangzhou 510640, China; li-yuan-fei@outlook.com (Y.L.); yumiao@gdaas.cn (M.Y.); tianzhimei@gdaas.cn (Z.T.); 3Key Laboratory of Animal Nutrition and Feed Science in South China, Ministry of Agriculture and Rural Affairs, Guangzhou 510640, China; 4Guangdong Provincial Key Laboratory of Animal Breeding and Nutrition, Institute of Animal Science, Guangdong Academy of Agricultural Sciences, Guangzhou 510640, China; 5Guangdong Engineering Technology Research Center of Animal Meat Quality and Safety Control and Evaluation, Institute of Animal Science, Guangdong Academy of Agricultural Sciences, Guangzhou 510640, China; 6Institute of Biological Technology, Nanchang Normal University, Nanchang 330032, China

**Keywords:** magnolol, polyphenol, amino acid, arginine uptake, immune, pigs

## Abstract

This study investigated whether dietary supplementation with magnolol affects growth performance, anti-inflammatory abilities, serum and muscle amino acid profiles, and metabolisms in growing pigs. A total of 42 seventy-days-old growing barrows (Duroc × Landrace × Yorkshire) were randomly allocated into two dietary groups: Con, control group (basal diet); and Mag, magnolol group (basal diet supplemented with 400 mg/kg of magnolol). The results revealed that dietary supplementation with magnolol had no effect (*p* > 0.05) on growth performance. However, magnolol supplementation remarkably increased (*p* < 0.05) the serum content of albumin, total protein, immunoglobulin G, immunoglobulin M, and interleukin-22. In addition, dietary magnolol supplementation altered the amino acid (AA) profiles in serum and dorsal muscle and particularly increased (*p* < 0.05) the serum content of arginine and muscle glutamate. Simultaneously, the mRNA expression of genes associated with AA transport in jejunum (*SLC38A2*, *SLC1A5*, and *SLC7A1*) and ileum (*SLC1A5* and *SLC7A1*) was higher (*p* < 0.05) in the Mag group than in the Con group. Additionally, the serum metabolomics analysis showed that the addition of magnolol significantly enhanced (*p* < 0.05) arginine biosynthesis, as well as *D*-glutamine and *D*-glutamate metabolism. Overall, these results suggested that dietary supplementation with magnolol has the potential to improve the accumulation of AAs, protein synthesis, immunity, and body health in growing pigs by increasing intestinal absorption and the transport of AAs.

## 1. Introduction

In recent decades, the use of subtherapeutic doses of antibiotics over extended periods in animal production has been identified as a major contributor to the rapid dissemination of antimicrobial resistance [1]. Although some countries have bans on antibiotics in livestock feed [2], these prohibitions have adversely affected productivity within the realm of livestock farming [3,4]. As a result, there is an urgent need for alternative approaches to animal nutrition that do not rely on antibiotics. Plant extracts [5,6], probiotics [7], and antimicrobial peptides [8] all represent potential alternatives to antibiotics for growth promotion in livestock. Specifically, polyphenols gained particular attention due to their ability to serve as effective substitutes for antibiotics, while, at the same time, enhancing systemic immunity and restoring intestinal barrier function [9,10].

The rapid growth of skeletal muscle is a prominent characteristic in growing pigs that is heavily reliant on the availability of amino acids (AAs) for protein synthesis [11]. AA absorption mainly occurs in the small intestine [12], and although antibiotics can enhance their transport and absorption by reducing intestinal-wall thickness [13] and upregulating the expression of genes related to this process [14], they also disrupt gut microbiota and immune homeostasis, thus leading to antimicrobial resistance [15].

In recent years, the beneficial effects of polyphenols on growth promotion [16], intestinal barrier functions [17], and the uptake of AAs in vivo and in vitro [16,18,19,20] have been confirmed. For instance, Chen et al. [21] demonstrated that caffeic acid can improve intestinal barrier function by reducing inflammation in weaning piglets. Similarly, gallic acid was found to enhance the uptake of major types of AAs by upregulating the expression of *SLC7A1* in intestinal porcine enterocyte J2 cell line and middle-jejunum segments [18]. Additionally, dietary supplementation with 400 mg/kg of chlorogenic acid was shown to improve AA profiles and muscle protein biosynthesis in finishing pigs [16]. However, it remains unclear whether all the different polyphenols have the capacity to enhance AA absorption, and, thus, further research is necessary.

Magnolol, a natural and nontoxic polyphenolic compound found in the bark of the Chinese herbal medicine *Magnolia officinalis*, has a wide range of pharmacological effects on humans and animals [22]. Recently, magnolol has emerged as an effective and safe feed additive to enhance health and growth in livestock production. Dietary magnolol supplementation in chickens has been reported to enhance growth performance [23], exert anti-inflammatory and antioxidant effects [24,25], increase villus height [26], promote intestinal barrier function [27], and regulate glucose and lipid metabolism [28]. Despite sharing common characteristics with other polyphenols, the potential of magnolol to regulate AA transporters and thereby enhance the absorption or deposition of AAs to promote animal growth remains uncertain.

Findings from our previous study supported the growth-promoting effects of magnolol and its ability to enhance the apparent digestibility of crude protein in weaning piglets [29]. Therefore, we postulated that magnolol supplementation may optimize AA profiles, stimulate protein synthesis, and facilitate overall growth and development. The present study aimed to determine the effects of dietary magnolol supplementation on growth performance, serum biochemical parameters, serum, and muscle AA profiles, as well as the relative mRNA expression of AA transporters in growing pigs. The outcomes derived from this will contribute to establishing a solid theoretical basis for future applications within the domain of pig production.

## 2. Results

### 2.1. Growth Performance

The growth performance of growing pigs in different diet groups is summarized in Table 1. The average daily gain (ADG) of growing pigs showed no significant difference between the control (Con) and magnolol (Mag) groups (*p* > 0.05). Similarly, no significant differences in the average daily feed intake (ADFI) and feed conversion rate (F:G) were observed between the two groups (*p* > 0.05).

### 2.2. Serum Biochemical Indices

The serum biochemical indices of pigs fed a magnolol-supplemented diet are shown in Table 2. The serum levels of total protein (TP) and albumin (ALB) in the Mag group were significantly higher than those in the Con group (*p* < 0.05). The serum level of high-density lipoprotein cholesterol (HDL-C) tended to increase with magnolol supplementation (*p* < 0.10). The concentration of diamine oxidase (DAO) showed a decreasing trend in response to magnolol supplementation (*p* < 0.10). However, there was no significant difference in the other serum biochemical indices (low-density lipoprotein cholesterol, LDL-C; alanine transaminase, ALT; aspartate transaminase, AST; glucose, Glu; urea; and uric acid, UA) between the two groups (*p* > 0.05).

### 2.3. Immune Responses

The serum immunological markers were measured to evaluate the immune status of growing pigs (Figure 1). Concerning the serum immune response, dietary supplementation with magnolol significantly increased the cytokine levels of immunoglobulin G (IgG), immunoglobulin M (IgM), and interleukin-22 (IL-22) compared to those in the Con group (Figure 1B,C,F) (*p* < 0.05). There was a decreasing tendency shown by the interferon-gamma (IFN-γ) concentration in the Mag group in comparison to the Con group (Figure 1G) (*p* < 0.10). However, there were no significant differences in immunoglobulin A (IgA), interleukin-2 (IL-2), interleukin-10 (IL-10), and tumor necrosis factor-alpha (TNF-α) levels between the Con and Mag groups (Figure 1A,D,E,H) (*p* > 0.05).

### 2.4. Serum and Dorsal Muscle Amino Acid (AA) Profiles

The AA profiles of serum detected using high-performance liquid chromatography (HPLC) are shown in Table 3. Interestingly, the serum concentrations of arginine and proline in the Mag group were much higher than those in the Con group (*p* < 0.05). Additionally, dietary supplementation with magnolol showed an increasing trend in the serum concentrations of methionine, aspartate, and cysteine (*p* < 0.10). However, there were no marked changes in the serum concentrations of isoleucine, lysine, phenylalanine, threonine, tryptophan, valine, alanine, glutamate, glycine, serine, tyrosine, asparagine, glutamine, essential, nonessential, and total AAs between the Con and Mag groups (*p* > 0.05).

The AA profiles in the dorsal muscle, as determined using HPLC, are shown in Table 4. A fraction of the dorsal muscle AA concentrations was greatly influenced by magnolol supplementation. Concretely, the concentrations of glutamate and tyrosine in the Mag group were much higher than those in the Con group (*p* < 0.05). Moreover, pigs fed a magnolol-supplemented diet tended to have higher dorsal muscle histidine, leucine, methionine, alanine, cysteine, and total AA contents (*p* < 0.10). Conversely, no significant differences in dorsal muscle arginine, isoleucine, lysine, phenylalanine, threonine, tryptophan, valine, aspartate, glycine, serine, asparagine, glutamine, and proline contents were detected between the two groups (*p* > 0.05).

### 2.5. Gene Expression of AA Transporters in the Jejunum and Ileum

Given the changes influenced by magnolol supplementation in serum AA profiles and the importance of AA transporters in the regulation of AA transport, the mRNA expression levels of the genes associated with AA transporters in both jejunum and ileum were examined (Figure 2). In the jejunum (Figure 2A), the magnolol-supplemented diet sharply increased the relative mRNA expression levels of AA transporters, including solute carrier family 38 member 2 (*SLC38A2*/*SNAT2*), solute carrier family 1 member 5 (*SLC1A5/ASCT2*), and solute carrier family 7 member 1 (*SLC7A1/CAT1*), compared with the Con group (*p* < 0.05). However, there were no differences in the relative mRNA expression levels of AA transporters such as solute carrier family 7 member 9 (*SLC7A9*/*b^0,+^AT*), solute carrier family 3 member 1 (*SLC3A1/rBAT*), solute carrier family 1 member 3 (*SLC1A3/EAAT1*), solute carrier family 7 member 8 (*SLC7A8/LAT2*), solute carrier family 7 member 7 (*SLC7A7/y^+^LAT1*), and argininosuccinate synthase 1 (*ASS1*) between the Con and Mag groups (*p* > 0.05).

In the ileum (Figure 2B), the relative mRNA expression levels of *SLC1A5* and *SLC7A1* were markedly higher in the Mag group than those in the Con group (*p* < 0.05). However, dietary magnolol supplementation did not affect the relative mRNA expression levels of *SLC38A2*, *SLC7A9*, *SLC3A1*, *SLC1A3*, *SLC7A8*, *SLC7A7*, and *ASS1* (*p* > 0.05).

### 2.6. Serum Metabolic Profiles and Pathway Analysis

To further investigate the serum metabolic profiles in growing pigs fed a magnolol-supplemented diet (Figure 3), the untargeted serum metabolome was individually profiled using UPLC-Orbitrap-MS/MS and then compared between the two groups. In our study, a total of 161 metabolites were detected in each group, and the partial least squares–discriminant analysis (PLS-DA) (Figure 3A) showed a clear separation between the Con and Mag groups, indicating that magnolol supplementation induced an alteration in serum metabolic profiles in growing pigs.

Next, the response permutation test (RPT) models (Figure 3B) revealed that the PLS-DA models were highly precise and reliable. The variable selection method, using variable importance in the project (VIP) > 1 and *p* < 0.05, was used to identify the significantly different metabolites. In total, 20 metabolites with VIP > 1 and *p* < 0.05 were ultimately identified (Figure 3C and Appendix A). As shown in the Z-score heatmaps (Figure 3C), 4 metabolites (2-phenylacetamide, estrone-3-glucuronide, *L*-arginine, and *N*-a-acetyl-*L*-arginine) were enriched and 16 metabolites (2-hydroxyquinoline, 2-oxoglutaric acid, 2-phenylacetamide, 3-hydroxybenzoic acid, 4-nitrophenol, acetophenone, adipic acid, alpha-hydroxyhippuric acid, alpha-hydroxyisobutyric acid, creatine, creatinine, hippuric acid, jasmonic acid, N6, N6, N6-trimethyl-*L*-lysine, phenylacetylglycine, ribonolactone, and thymine) were decreased in the Mag group compared with the Con group.

Furthermore, a Kyoto Encyclopedia of Genes and Genomes (KEGG) enrichment analysis (Figure 4A) was performed to investigate the potential metabolic pathways mediating via the effect of magnolol. The KEGG pathway enrichment analysis identified 12 major metabolic pathways affected by dietary supplementation with magnolol, including AA metabolism (arginine biosynthesis; arginine and proline metabolism; *D*-glutamine and *D*-glutamate metabolism; phenylalanine metabolism; lysine degradation; alanine; aspartate and glutamate metabolism; and glycine, serine, and threonine metabolism), carbohydrate metabolism (butanoate metabolism and citrate cycle (tricarboxylic acid (TCA) cycle)), nucleotide metabolism (pyrimidine metabolism and aminoacyl-tRNA biosynthesis), and lipid metabolism (steroid hormone biosynthesis).

Through the analysis of key metabolites associated with the AA metabolic pathway, *L*-arginine was identified as the most crucial metabolite influencing AA metabolic pathway. A fold-change analysis obtained from the arginine concentration detection by UPLC-Orbitrap-MS/MS (Figure 4B) and HPLC (Figure 4C) was performed to bolster the robustness of the findings that the level of arginine was altered by magnolol supplementation. Magnolol supplementation significantly upregulated the levels of arginine, as observed in both UPLC-Orbitrap-MS/MS and HPLC analyses (*p* < 0.05), resulting in respective 2.42-fold and 1.59-fold increases.

## 3. Discussion

Ensuring adequate nutrition during the animal’s growth period is crucial in modern agriculture to maximize genetic potential for growth [11]. Insufficient supplies of AAs during the growth phase can lead to impaired growth and decreased skeletal muscle mass [30]. Polyphenols have recently attracted enormous attention due to their ability to facilitate the absorption and transport of AAs in the small intestine [18], as well as maintain intestinal health [17]. Our preliminary results demonstrated that magnolol, a potent herbal polyphenolic compound, was able to enhance the apparent digestibility of crude protein in pigs [29]. However, the mechanisms underlying the increased protein absorption and utilization induced by magnolol remain unclear. In this study, magnolol supplementation significantly enhanced serum TP, ALB, and immune factors, while also inducing alterations in both serum and muscle AA profiles. Furthermore, these alterations were associated with the upregulated expression of AA-transport-related genes and the enriched metabolic pathways involved in AA synthesis and metabolism. Taken together, magnolol has the potential to serve as a nutritional supplement to enhance the growth of growing pigs by increasing protein synthesis and muscle mass, while optimizing their genetic growth capacity.

### 3.1. Effects of Dietary Magnolol Supplementation on the Growth Performance of Growing Pigs

The assessment of the growth performance in animals provides critical evidence to support the use of magnolol as a feed additive in animal production. Although the beneficial effects of magnolol on poultry production and health have been extensively documented [23,24,26,31], there is limited research on its dietary effects in pigs. Our study found that the addition of magnolol to the diet of pigs did not produce significant effects on ADG, ADFI, or F:G in growing pigs. This study represents the first investigation into the impact of a magnolol-supplemented diet on growing pigs, while the results are not consistent with our previous study, which showed an enhanced growth performance in weaning pigs with a 400 mg/kg magnolol supplementation [29]. Previous studies showed that the incorporation of magnolol as an additive can have various effects on growth performance. In line with our findings, Menci et al. [32] reported no remarkable effect on growth performance in finishing pigs after dietary supplementation with magnolia bark extract (mainly containing magnolol). Similarly, Xie et al. [23] found that diets supplemented with 100–300 mg/kg of magnolol did not improve the growth performance of yellow-feathered broilers from day 1 to 28, but from day 28 to 51, groups receiving 200 and 300 mg/kg magnolol had higher ADG compared to the control group. Previous studies showed that the effects on growth performance were typically observed under conditions where the animals had difficulties in digesting the diet or when they were exposed to unsuitable environmental circumstances [33]. However, our study suggested that magnolol supplementation did not significantly promote growth performance. This may be ascribed to the potential allocation of additional energy derived from feed absorption in response to magnolol supplementation towards augmenting the immune system or other pathways, rather than promoting growth.

### 3.2. Effects of Dietary Magnolol Supplementation on Biochemical Indices of Growing Pigs

The serum biochemical indices serve as functional indicators that reflect the metabolic processes and nutrient deposition in animals [34]. Although no significant effect on growth performance was observed in our current studies, dietary supplementation with magnolol did have a fair impact on serum indices. We used specific biochemical indices related to protein and nitrogen metabolism to evaluate protein synthesis, turnover, and the physiological status of growing pigs through analysis of their serum. The liver is responsible for the production of most blood protein, and animals prefer AAs absorbed from the intestinal lumen as precursors for liver protein synthesis [35]. The TP level in blood serves as a key indicator of hepatic protein metabolic status in response to dietary supplementation. ALB, the most abundant blood protein, acts as an essential binding transporter for various metabolites, while also providing a constant supply of endogenous AAs [36]. Our current study detected a significant increase in serum TP and ALB levels as a result of dietary supplementation with magnolol. Previous relevant studies have reported similar findings, suggesting that the incorporation of 200 mg/kg and 300 mg/kg of magnolol effectively increased the concentration of serum TP in *Linwu* ducklings [31]. These studies suggested that the supplementation with magnolol may potentially enhance liver functions, thereby increasing AA transport and utilization.

HDL-C, which acts as a “lipid scavenger” in the bloodstream, not only promotes hepatic health and regulates cholesterol homeostasis but also participates in interactions with plasma proteins that have anti-inflammatory effects [37]. Our study revealed that dietary supplementation with magnolol resulted in a significant increase in HDL-C content compared to the Con group, consistent with Xie et al. [28] findings demonstrating the ability of magnolol to increase HDL-C production and maintain lipid metabolic homeostasis. Therefore, it is reasonable to suggest that the magnolol-induced increase in serum HDL-C levels may potentially enhance liver functions and ameliorate lipid metabolism.

DAO serves as a valuable blood biomarker to assess the integrity of the mechanical intestinal barrier, with increased levels indicating impaired mechanical barrier function. Moreover, the mechanical barrier function serves as the histological foundation that maintains intestinal barrier function and prevents the entry of harmful substances, such as bacteria and endotoxins, across the intestinal mucosa into bloodstream [38]. Dietary supplementation with magnolol may potentially enhance mechanical intestinal barrier function in growing pigs, as evidenced by the reduced DAO content detected in the Mag group compared to the Con group. These findings were similar to those of previous research conducted by Deng et al. [39], suggesting that magnolol improves mechanical intestinal barrier integrity. In conclusion, our studies indicated that dietary supplementation with magnolol enhances liver function and improves mechanical intestinal barrier integrity in growing pigs, thereby facilitating the absorption and utilization of AAs.

### 3.3. Effect of Dietary Magnolol Supplementation on the Immune Response of Growing Pigs

An increase in serum immunoglobulin levels is a clear indication of the enhanced innate immunity of animals, with IgG and IgM being the primary immunoglobulins that protect against pathogen invasion into the bloodstream and promote disease resistance [40]. Our findings corroborated studies conducted by Ding et al. [41] in broiler chickens and Huang et al. [42] in Nile tilapia (*Oreochromis niloticus*), demonstrating that dietary supplementation with magnolol significantly increases serum levels of IgG and IgM. Previous studies have also demonstrated that IgM production is dependent on ALB as its primary substrate [43], and an increase in both ALB and TP levels has been associated with enhanced innate immunity [44,45]. Therefore, it is reasonable to hypothesize that the observed increase in IgG and IgM levels following magnolol treatment may be partially attributed to the concurrent elevation of blood TP and ALB levels as a result of magnolol supplementation.

The cytokine IL-22, which is involved in tissue signaling, plays a pivotal role in hepatoprotection, tissue proliferation, and repair. Additionally, it also serves as a protective barrier against bacterial-induced damage and inflammation of the intestinal epithelium, while preserving the integrity of the intestinal barrier [46,47]. Prior to this study, limited research had been published on the impact of in-feed magnolol on the regulation of IL-22 secretion. Our findings revealed a significant increase in serum IL-22 levels by dietary magnolol supplementation. Furthermore, our unpublished results support these results by revealing a substantial increase in serum IL-22 levels among weaning pigs. Thus, the promotion of liver health and enhancement of intestinal barrier function, mediated by the secretion of magnolol-induced IL-22, can partially account for its effect on increasing serum TP and ALB levels.

Similarly, IFN-γ, a proinflammatory cytokine, can enhance the ongoing immune response and facilitate apoptosis induction. The level of IFN-γ was modulated by IL-22, which exerted antagonistic effects on its activity [48]. Therefore, this study measured the level of IFN-γ in serum and found a reduction in its content upon dietary supplementation with magnolol. Noteworthy, experimental findings by Wang et al. [49] indicated that honokiol treatment reduced serum IFN-γ levels by inhibiting the NF-κB signaling pathway. Moreover, previous studies have demonstrated that honokiol and magnolol, which are structural isomers of each other, exhibited similar inhibitory effects on the NF-κB signaling pathway in reducing inflammation factors [50]. Therefore, it is reasonable to assume that magnolol may also inhibit the NF-κB signaling pathway, leading to a reduction in the IFN-γ level. Overall, the results of this study demonstrated that the inclusion of dietary magnolol in the diet of growing pigs can effectively enhance their innate immunity, thereby contributing to enhancing the abilities to resist the diseases and microbial invasion.

### 3.4. Effect of Dietary Magnolol Supplementation on the Absorption and Utilization of AAs in Growing Pigs

AAs not only serve as the fundamental building blocks for protein synthesis but also play pivotal roles in both development and the immune response [51]. Blood functions as the primary circulating reservoir of AAs, while serum AAs act as metabolic intermediates for both protein nitrogenous anabolism and catabolism [12]. Our findings demonstrated that pigs fed with diets containing magnolol increased the levels of the majority of serum AAs compared to those in the Con group, which is analogous to a previous study by Wang et al. [16], indicating that the polyphenol chlorogenic acid could increase both serum and muscle AA contents. The levels of AAs in blood were mainly determined by their absorption through a specific transporter system in the small intestine [12]. We also found that dietary supplementation with magnolol led to an increase in serum AAs levels, concomitant with the increase in the mRNA expression of genes associated with AA transport in both the porcine jejunum and ileum. Therefore, a potential explanation is that the observed increase in serum AA content following magnolol treatment may be attributed to the upregulation of AA transporters, resulting in enhanced absorption of AA into the bloodstream. Interestingly, HPLC analysis (absolute quantification, Figure 4C) revealed that dietary magnolol supplementation significantly increased serum arginine levels by approximately 59%, which was also demonstrated by a corresponding significant increased serum arginine levels obtained from UPLC-Orbitrap-MS/MS analysis (relative quantification, Figure 4B). Recent evidence suggested that arginine is a crucial component in protein deposition and the formation of muscle fibers, while also producing nitric oxide and polyamines via metabolism, ultimately promoting cell proliferation and immunonutrition [52,53]. Therefore, dietary magnolol supplementation led to a significant increase in serum arginine levels, which can partially explain the observed enhancements in serum TP, ALB, and immune factors such as IgM, IgG, and IL-22.

Skeletal muscles, which serve as the largest reservoir of AAs [54], can effectively reflect the impact of dietary magnolol supplementation on AA turnover and deposition. The increase in the content of AAs in the skeletal muscle occurred concomitantly with the increase in the relative content of AAs in the serum (Table 3 and Table 4). Previous studies suggested that AA transporters have dual functions as both transporters and receptors, enabling them to detect changes in the cellular AA pool and external AAs, thereby enhancing nutrient signaling [55,56]. Thus, our studies revealing that dietary magnolol supplementation concurrently increased AA levels in serum and in muscle suggested that magnolol treatment may increase the AA availability to the host. Taken together, the increased levels of AAs in the dorsal muscle of growing pigs as a result of dietary magnolol supplementation may contribute to their muscle growth and mass accumulation. Notably, we detected a substantial increase of approximately 61% in muscle glutamate content. Glutamate, a substance centrally posited in mammalian metabolism, plays a crucial role in tissue synthesis, particularly in muscle, where it promotes intramuscular protein synthesis and facilitates a rapid gain of lean tissues in growing pigs [57,58,59]. Reeds et al. [60] reported that dietary glutamate had no significant impact on muscle glutamate content since it was mainly utilized by intestinal epithelium instead of being absorbed into the portal vein. Moreover, the body predominantly relies on de novo synthesis for the production of glutamate. However, arginine can be catabolized to generate various compounds, such as glutamate and glutamine [61]. Increased levels of arginine in the bloodstream may result in an increase in muscle glutamate content [62,63]. These studies aligned with our study, where a significant increase in serum arginine levels, no change in serum glutamate levels, no altering in muscle arginine levels, and a significant elevation in muscle glutamate levels were observed in response to magnolol supplementation. Thus, the inclusion of magnolol in the pig diets led to increased levels of muscle glutamate, potentially due to its ability to increase the serum arginine concentration and subsequently promote glutamate accumulation in muscle tissue. However, further research is necessary to confirm this hypothesis.

As secondary transporters, AA transporters facilitated the transfer of AAs [64]. To investigate the impact of magnolol supplementation on serum AAs, we examined the gene expression of AA transporters involved in intestinal AA absorption. Our findings revealed that the relative expression of mRNA levels of *SLC38A2*, *SLC1A5*, and *SLC7A1* in the jejunum was significantly higher in the Mag group than in the Con group. Additionally, the relative expressions of mRNA levels of *SLC1A5* and *SLC7A1* were also significantly higher in the ileum of pigs fed a magnolol-supplemented diet. Alanine–serine–cysteine amino acid transporter 2, encoded by *SLC1A5*, is capable of transporting a wide range of AAs, particularly methionine and proline in conjunction with the sodium-dependent neutral amino acid transporter 2, which is encoded by the *SLC38A2* gene [65,66]. Additionally, the cationic amino acid transporter 1, encoded by *SLC7A1*, serves as an AA transporter mainly responsible for the transport of arginine, histidine, and lysine [67]. The increased expression of the mRNA levels of these specific transporters observed in growing pigs fed a magnolol-supplemented diet provided compelling evidence for the significantly higher serum AA concentrations observed in the Mag group compared to the Con group. Earlier studies demonstrated that dietary supplementation with magnolol can enhance villus height in the jejunum and ileum, thereby facilitating increased AA uptake [26,68]. Therefore, it was plausible to speculate that the enhancement of AA uptake can be attributed to the effects of magnolol treatment in promoting intestinal development and increasing the relative expression of mRNAs of AA transporters.

Metabolomics is an effective tool to determine alterations in various metabolic pathways and biomarkers [69]. Our studies found significant differences in serum profiles between the Con and Mag groups, using the PLS-DA model. The KEGG enrichment analysis revealed that different affected metabolic pathways were primarily related to arginine biosynthesis, arginine and proline metabolism, and *D*-glutamine and *D*-glutamate metabolism. Previous studies in mice showed that dietary supplementation with magnolol mainly regulated metabolic pathways associated with arginine and proline metabolism; alanine, aspartate, and glutamate metabolism; and tryptophan metabolism [70,71], and such findings were akin to our findings. These findings suggested that dietary magnolol supplementation primarily affected the metabolism of compounds related to serum AAs, which may explain the conversion of arginine to glutamate in muscle following magnolol treatment through alterations in metabolic pathways regulating this process.

## 4. Materials and Methods

### 4.1. Animal, Diet, and Experimental Design

A total of 42 healthy 70-day-old crossbred growing barrows (Duroc × Landrace × Large White) were randomly divided into 2 groups according to their body weight (the average BW in the control and magnolol groups were 25.32 and 25.31 kg, respectively). Each group consisted of 7 replicates, with 3 pigs per replicate (pen). Pigs in 2 groups were fed the basal diet (CON) and the basal diet supplemented with 400 mg/kg magnolol (Mag), respectively. The experiment lasted for 35 days, including a 5-day adaptation period and a 30-day experimental period. The reared trial was selected from June to July at the Institute of Animal Science, Guangdong Academy of Agricultural Sciences, China. During the feeding experiment, the temperature was not controlled. The ambient temperature was within the range of 25–31 °C, and the humidity was approximately 65%, with a 13-hour natural light and 11-hour dark cycle. There were 14 pens in total, and all pens were located in one building. Pigs resided in adjacent individual concrete pens (4.5 m × 1.6 m) with concrete floors and catered with nipple drinkers and manual metal feeders and allowed ad libitum access to the diet and fresh water throughout the trial. The basal diet (Appendix A) meets the nutritional requirements for NRC 2012. Health conditions and feed consumption of pigs in each replicate were recorded weekly. Magnolol (purity, ≥98%) was obtained from Hunan Zhongmao Biological Technology Co., Ltd. (Changsha, China).

### 4.2. Growth Performance

Each pig was weighed on day 1 (initial BW) and day 35 (final BW) after fasting for 6 h to calculate the ADG (per pig). The feed intake of pigs was recorded for each pen weekly to calculate the ADFI (per pig), which was calculated based on the total feed intake per pen divided by the number of trial days (35 days) and the number of pigs in each pen (3 pigs). The F:G was computed as the average of feed intake divided by BW gain in each replicate.

### 4.3. Sample Collection

At the end of the trial, a fasting period of 6 h was implemented for all pigs, and one pig with approximately average BW (each group) was selected from each replicate. Blood was collected from the jugular vein, using non-heparin-coated vacutainer tubes, and centrifuged and separated to obtain serum. Pigs were euthanized by electrical stunning, followed by exsanguination, and the tissue of the mid-jejunum and mid-ileum was collected and placed into sterile Eppendorf tubes. Then, all samples were stored at −80 °C until measurement for serum biochemical indices, antioxidant capacity, inflammatory cytokines, AA profiles, and metabolome analysis.

### 4.4. Serum Biochemistry and Immunological Indices

The basic serum biochemical indices, including TP, LDL-C, HDL-C, ALT, AST, ALB, Glu, urea, UA, and DAO, were analyzed using a Selectra XL Autoanalyzer (Vita Scientific NV, Dieren, The Netherlands). The assay kits were purchased from Biosino Bio-Tec (Beijing, China). Furthermore, the levels of TNF-α, IFN-γ, IgG, IgA, IgM, IL-2, IL-10, and IL-22 were measured using commercial ELISA kits (North Institute of Biological Technology Co., Ltd., Beijing, China).

### 4.5. Free Amino Acids

The serum and dorsal muscle samples were prepared according to a previously described method [72]. Approximately 0.3 mL of serum or 0.3 g of muscle was homogenized with 1 mL sulfosalicylic acid (20% *w*/*v*) and centrifuged (12,000× *g*, 15 min, 4 °C). After filtering the supernatant with a 0.22 μm filter, a HPLC-*L*-8900 Amino Acid Analyzer (Hitachi Ltd., Tokyo, Japan) was used to measure the AA levels. Essential AAs included histidine, leucine, isoleucine, lysine, methionine, phenylalanine, threonine, valine, arginine, and tryptophan; and nonessential AAs consisted of alanine, aspartate, glutamate, glutamine, glycine, serine, tyrosine, cysteine, and asparagine.

### 4.6. Quantitative Real-Time Polymerase Chain Reaction (qRT-PCR)

Total RNA was extracted from jejunal and ileum tissue homogenates, using the TRIzol reagent (Bioshape, Hefei, China). A Nanodrop 1000 spectrophotometer (Thermo Fisher Scientific Inc., Waltham, MA, USA) was used to measure the concentrations of total RNA. Subsequently, the synthesis of cDNA was performed using a cDNA Synthesis Kit (Takara Biotechnology, Dalian, China). Primers of selected genes are described in Appendix A. The mRNA abundance was measured using a CFX96 Real-Time PCR Detection System (Bio-Rad Laboratories Inc., Hercules, CA, USA). The reaction conditions were set according to those of a previous study [14]. The relative expression of each gene was calculated according to the following formula: 2^−(∆∆Ct)^ [73], where ∆∆Ct = (Ct_target_ − Ct_β−actin_)_treatment_ − (Ct_target_ − Ct_β−actin_)_control_, and β-actin served as the reference housekeeping gene.

### 4.7. Untargeted Serum Metabolomics Analysis

The serum metabolite samples were pretreated and extracted, referring to previous studies [68,74], but with minor modifications. In brief, methanol was added to the serum sample to remove all serum proteins. Then, the centrifuged supernatant was blow-dried with nitrogen for 2 h and processed. The quality-control (QC) sample was mixed with 14 serum samples to assess the stability. A Dionex UltiMate 3000 UPLC system coupled with a high-resolution Q-Exactive Focus mass spectrometer (UPLC-Orbitrap-MS/MS, Thermo Fisher Scientific, Waltham, MA, USA) was utilized to detect the serum metabolic profiles [74]. The Compound Discover 2.1 (Thermo Fisher Scientific, Waltham, MA, USA) data-analysis tool was utilized to preprocess the raw data and identify the metabolites, using the mzCloud and mzVault libraries. Additionally, the PLS-DA of metabolites was performed with the SIMCA-P (Version 14.1, Sartorius Stedim Data Analytics AB, Umeå, Sweden), and the RPT model was performed to evaluate the accuracy of the PLS-DA model. In addition, the VIP was computed in the PLS-DA model, and the differential metabolites were acquired by selecting the metabolites that comply with the VIP > 1 and a *p* < 0.05 criteria. To make a further exploration of metabolic changes, the differential metabolites were analyzed on MetaboAnalyst 5.0 (https://www.metaboanalyst.ca/, accessed on 19 March 2023), utilizing the KEGG pathway enrichment analysis.

### 4.8. Statistical Analysis

The experimental raw data were processed using Microsoft Excel 2019. Then, the SPSS 26.0 software (IBM Corporation, Armonk, NY, USA) was used to perform an independent sample *t*-test, and GraphPad Prism 8.0.3 software (GraphPad Software Inc., San Diego, CA, USA) was used for graphical representation. All of the data are expressed as the mean ± standard error of the mean (SEM). Statistical significance was based on *p* < 0.05, and the trend of difference was considered to be 0.05 < *p* ≤ 0.10.

## 5. Conclusions

The current study suggested that dietary supplementation with magnolol was able to enhance the absorption and utilization of AAs in growing pigs by upregulating the gene expression of key small intestinal AA transporters, promoting anti-inflammatory ability and potentially contributing to the improvement of intestinal barrier functions. The metabolomic analysis and KEGG enrichment analysis further showed that magnolol regulates AA metabolism pathways to promote skeletal muscle growth and development. However, the information regarding the biomarker of intestinal barrier functions in response to magnolol supplementation and the mechanisms underlying the effect of magnolol on the conversion of arginine to glutamate was limited. This study highlights the underlying mechanism by which magnolol exerted its effects on protein production, providing a potential alternative for harnessing the growth potential of pigs.

## Figures and Tables

**Figure 1 ijms-24-13952-f001:**
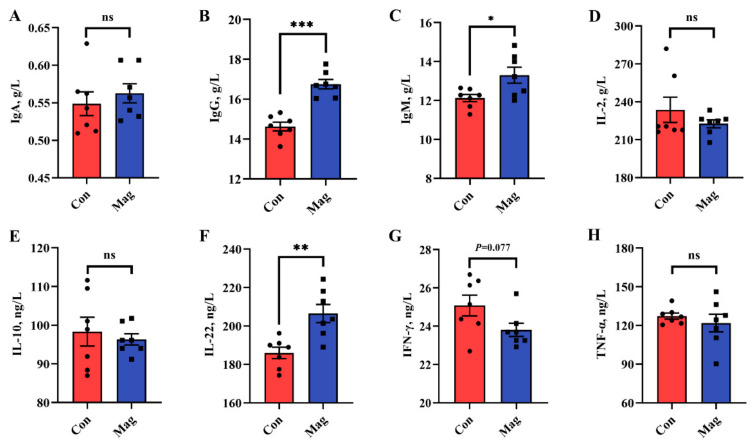
Effect of dietary supplementation with magnolol on immune responses of growing pigs. (**A**) Immunoglobulin (Ig) A, (**B**) IgG, (**C**) IgM, (**D**) interleukin (IL)-2, (**E**) IL-10, (**F**) IL-22, (**G**) interferon-gamma (IFN-γ), and (**H**) tumor necrosis factor-alpha (TNF-α). The results are presented as means ± SEM (*n* = 7). Con, control group, pigs fed a basal diet; Mag, magnolol group, pigs fed a basal diet supplemented with 400 mg/kg magnolol. The ns represents *p* > 0.05, * represents *p* < 0.05, ** represents *p* < 0.01, and *** represents *p* < 0.001.

**Figure 2 ijms-24-13952-f002:**
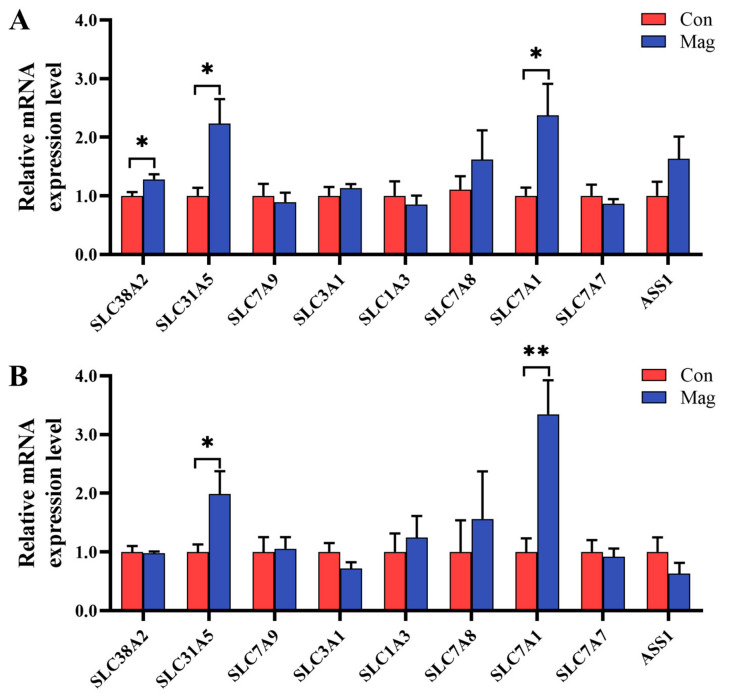
Effect of supplementation with magnolol on the mRNA expression levels of select AA transporters compared with growing pigs fed the control diet in the small intestine: (**A**) jejunum and (**B**) ileum. Con, control group, pigs fed a basal diet; Mag, magnolol group, pigs fed a basal diet supplemented with 400 mg/kg magnolol. β-actin was used as an internal control. *SLC38A2* (*SNAT2*), solute carrier family 38 member 2; *SLC1A5* (*ASCT2*), solute carrier family 1 member 5; *SLC7A9* (*b^0,+^AT*), solute carrier family 7 member 9; *SLC3A1* (*rBAT*), solute carrier family 3 member 1; *SLC1A3* (*EAAT1*), solute carrier family 1 member 3; *SLC7A8* (*LAT2*), solute carrier family 7 member 8; *SLC7A1* (*CAT1*), solute carrier family 7 member 1; *SLC7A7* (*y^+^LAT1*), solute carrier family 7 member and *ASS1*, argininosuccinate synthetase 1. * Represents *p* < 0.05 and ** represents *p* < 0.01. Values are presented as mean ± SEM (*n* = 7).

**Figure 3 ijms-24-13952-f003:**
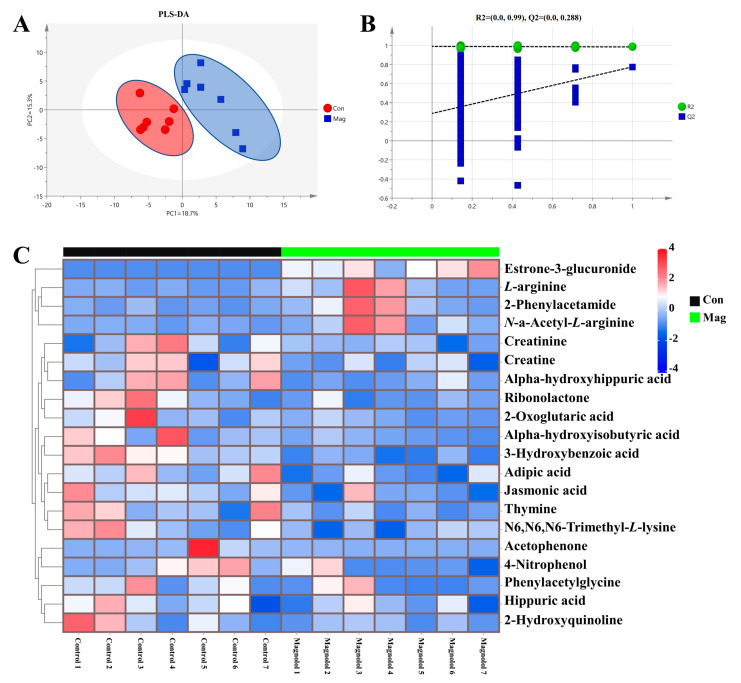
Effect of supplementation with magnolol on serum metabolic profiles of growing pigs. (**A**) Partial least squares-discriminant analysis, PLS-DA; (**B**) response permutation test, RPT; and the (**C**) Z-score heatmap of visualizing the remarkably changed metabolites (Z-score was utilized to transform the metabolites peak area) between the Con and the Mag group. Con, control group, pigs fed a basal diet; Mag, magnolol group, pigs fed a basal diet supplemented with 400 mg/kg magnolol. Values are presented as means ± SEM (*n* = 7).

**Figure 4 ijms-24-13952-f004:**
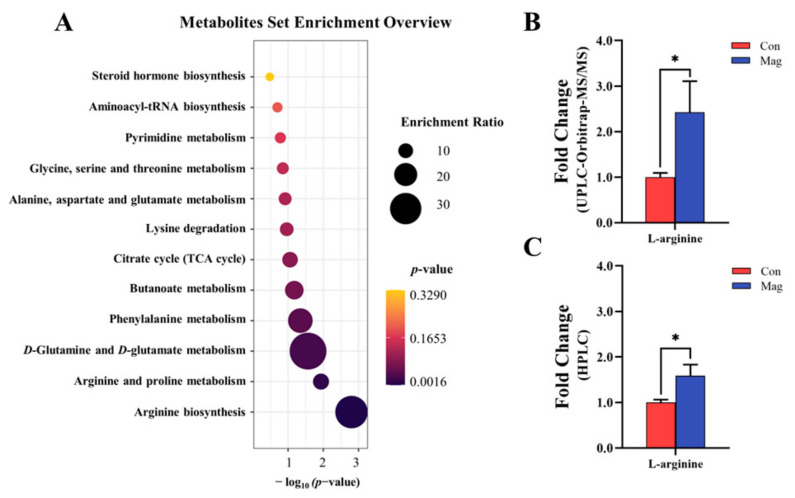
Effect of supplementation with magnolol on serum metabolic profiles of growing pigs. (**A**) KEGG metabolic pathways enrichment analysis. (**B**) The fold change of serum *L*-arginine level (the peak area) based on UPLC-Orbitrap-MS/MS (relative quantification). (**C**) The fold change of serum *L*-arginine level (the concentration) based on HPLC (absolute quantification). Con, control group, pigs fed a basal diet; Mag, magnolol group, pigs fed a basal diet supplemented with 400 mg/kg magnolol. * Represents *p* < 0.05, and values are presented as means ± SEM (*n* = 7).

**Table 1 ijms-24-13952-t001:** Effect of dietary supplementation with magnolol on the growth performance of growing pigs.

Items ^1^	Con ^2^	Mag ^3^	*p*-Value
Initial BW (kg) (day 0)	25.32 ± 0.04	25.31 ± 0.02	0.841
final BW (kg) (day 35)	50.00 ± 1.03	51.30 ± 0.76	0.334
ADFI, g/d	1526.14 ± 19.42	1521.79 ± 40.50	0.907
ADG, g/d	719.52 ± 18.04	742.31 ± 21.67	0.444
F:G	2.13 ± 0.04	2.05 ± 0.03	0.132

^1^ ADFI, average daily feed intake; ADG, average daily gain; BW, body weight; F:G, feed conversion rate; ^2^ Con, control group, pigs fed a basal diet; ^3^ Mag, magnolol group, pigs fed a basal diet supplemented with 400 mg/kg magnolol. Values are presented as means ± Standard Error of the Mean (SEM) (*n* = 7).

**Table 2 ijms-24-13952-t002:** Effect of dietary supplementation with magnolol on serum biochemical indices of growing pigs.

Items ^1^	Con ^2^	Mag ^3^	*p*-Value
TP, g/L	60.86 ± 1.62	65.79 ± 1.28 *	0.049
LDL-C, mmol/L	0.86 ± 0.03	0.81 ± 0.05	0.394
HDL-C, mmol/L	0.73 ± 0.03	0.83 ± 0.03	0.055
ALT, U/L	47.29 ± 8.14	32.82 ± 2.75	0.161
AST, U/L	78.33 ± 18.94	41.99 ± 3.77	0.107
ALB, g/L	19.61 ± 0.71	22.41 ± 0.81 *	0.033
GLU, mmol/L	4.83 ± 0.28	5.07 ± 0.22	0.530
UREA, mmol/L	7.74 ± 0.44	7.81 ± 0.75	0.937
UA, μmol/L	17.29 ± 1.65	15.66 ±1.99	0.592
DAO, U/L	14.47 ± 1.86	9.98 ± 0.68	0.057

^1^ ALB, albumin; ALT, alanine aminotransferase; AST, aspartate aminotransferase; GLU, glucose; DAO, diamine oxidase; HDL-C, high-density lipoprotein cholesterol; LDL-C, low-density lipoprotein cholesterol; TP, total protein. ^2^ Con, control group, pigs fed a basal diet; ^3^ Mag, magnolol group, pigs fed a basal diet supplemented with 400 mg/kg magnolol. Values are presented as means ± SEM (*n* = 7), and * represents *p* < 0.05.

**Table 3 ijms-24-13952-t003:** Effect of dietary supplementation with magnolol on AA profiles of growing pigs in serum (nmol/μL).

Items	Con ^1^	Mag ^2^	*p*-Value
Arginine	122.40 ± 6.86	194.53 ± 27.29 *	0.035
Histidine	56.35 ± 2.75	93.53 ± 22.60	0.180
Isoleucine	72.55 ± 7.18	92.80 ± 15.17	0.286
Leucine	153.83 ± 10.72	208.17 ± 25.43	0.101
Lysine	136.24 ± 9.01	205.28 ± 50.06	0.233
Methionine	112.29 ± 5.38	129.03 ± 5.61	0.070
Phenylalanine	73.04 ± 5.33	111.25 ± 26.30	0.232
Threonine	109.64 ± 5.51	133.00 ± 28.01	0.463
Tryptophan	42.69 ± 7.31	65.17 ± 19.92	0.357
Valine	226.16 ± 6.28	254.21 ± 18.24	0.212
Alanine	553.29 ± 56.70	618.18 ± 55.30	0.463
Aspartate	34.02 ± 5.76	56.09 ± 8.60	0.075
Glutamate	309.12 ± 45.53	331.30 ± 28.19	0.709
Glycine	853.82 ± 91.07	1081.41 ± 35.42	0.120
Serine	119.25 ± 10.83	147.63 ± 13.76	0.161
Tyrosine	63.00 ± 5.26	94.68 ± 17.57	0.136
Asparagine	50.39 ± 6.13	81.86 ± 17.38	0.140
Glutamine	402.68 ± 21.93	483.32 ± 43.07	0.148
Proline	249.14 ± 25.20	353.71 ± 35.42 *	0.048
Cysteine	83.26 ± 5.25	96.85 ± 2.82	0.063
Essential Amino Acids ^3^	1105.20 ± 29.84	1339.67 ± 130.17	0.136
Nonessential Amino Acids ^4^	2468.83 ± 203.00	2991.32 ± 234.98	0.146
Total Amino Acids ^5^	3574.03 ± 217.64	4709.96 ± 575.24	0.127

^1^ Con, control group, pigs fed a basal diet; ^2^ Mag, magnolol group, pigs fed a basal diet supplemented with 400 mg/kg magnolol. ^3^ Essential amino acids included arginine, histidine, isoleucine, leucine, lysine, methionine, phenylalanine, threonine, tryptophan, and valine. ^4^ Nonessential amino acids consisted of alanine, asparagine, aspartate, cysteine, glutamate, glutamine, glycine, serine, and tyrosine. ^5^ Total amino acids included essential amino acids, nonessential amino acids, and proline. The results are presented as means ± SEM (*n* = 7), and * represented *p* < 0.05.

**Table 4 ijms-24-13952-t004:** Effect of supplementation with magnolol on AA profiles of growing pigs in dorsal muscle (mg/100 g).

Items	Con ^1^	Mag ^2^	*p*-Value
Arginine	1.81 ± 0.21	2.60 ± 0.47	0.193
Histidine	0.76 ± 0.05	1.10 ± 0.17	0.098
Isoleucine	1.16 ± 0.12	1.33 ± 0.14	0.414
Leucine	2.53 ± 0.19	3.93 ± 0.65	0.099
Lysine	2.01 ± 0.31	2.64 ± 0.63	0.426
Methionine	1.95 ± 0.19	2.43 ± 0.09	0.067
Phenylalanine	2.37 ± 0.19	2.77 ± 0.32	0.339
Threonine	1.42 ± 0.15	1.81 ± 0.22	0.205
Tryptophan	0.65 ± 0.06	0.75 ± 0.04	0.231
Valine	1.92 ± 0.05	2.89± 0.77	0.288
Alanine	14.45 ± 0.78	17.06 ± 0.93	0.070
Aspartate	2.14 ± 0.21	1.98 ± 0.33	0.709
Glutamate	4.71 ± 0.34	7.62 ± 0.88 *	0.015
Glycine	14.98 ± 1.38	13.59 ± 1.09	0.481
Serine	2.20 ± 0.16	2.58 ± 0.18	0.170
Tyrosine	2.01 ± 0.12	2.64 ± 0.20 *	0.031
Asparagine	0.96 ± 0.08	1.35 ± 0.23	0.173
Glutamine	21.29 ± 1.71	21.32 ± 1.53	0.990
Proline	4.26 ± 0.57	4.31 ± 0.32	0.953
Cysteine	0.25 ± 0.03	0.35 ± 0.04	0.055
Essential Amino Acids ^3^	18.03 ± 0.88	21.23 ± 2.81	0.348
Nonessential Amino Acids ^4^	62.17 ± 4.44	71.55 ± 3.69	0.159
Total Amino Acids ^5^	78.80 ± 4.23	97.60 ± 6.69	0.052

^1^ Con, control group, pigs fed a basal diet; ^2^ Mag, magnolol group, pigs fed a basal diet supplemented with 400 mg/kg magnolol. ^3^ Essential amino acids included arginine, histidine, isoleucine, leucine, lysine, methionine, phenylalanine, threonine, tryptophan, and valine. ^4^ Nonessential amino acids consisted of alanine, asparagine, aspartate, cysteine, glutamate, glutamine, glycine, serine, and tyrosine. ^5^ Total amino acids included essential amino acids, nonessential amino acids, and proline. Values are presented as means ± SEM (*n* = 7); * represents *p* < 0.05.

## Data Availability

All relevant data are provided in the manuscript.

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
