# Peer review of "Magnolol Supplementation Alters Serum Parameters, Immune Homeostasis, Amino Acid Profiles, and Gene Expression of Amino Acid Transporters in Growing Pigs"

_ijms, 2023, doi:10.3390/ijms241813952_

Round 1
Reviewer 1 Report
Article
Dietary supplementation with magnolol alters serum biochemical indices, immune homeostasis, amino acid profiles and gene expression of amino acid transporter in the small intestine of growing pigs
Overview
The availability of dietary amino acids determinates the fast growth of skeletal muscle in piglets; magnolol- a polyphenol isolated from Magnolia officinalis bark that has been found to promote the broiler growing- may be a sustainable strategy to avoid the use of subtherapeutic dose of antibiotics used for animal production that contributes to microbial resistance. In this study, the effect of dietary magnolol was tested in piglets. Findings suggested that the regulatory effect of magnolol on the circulant amino amino acids, the mRNA expression of amino acid transporters, metabolomics and intestinal gut barrier biomarkers may provide novel pathways for animal production in porcine industry.
Highlights
Study of high impact on animal production in the porcine industry and very well written.
Major comments
The referee suggests that conclusions should be limited on data and include the limitations of this interesting contribution. In conclusions (line 512) authors claim that magnolol improved the intestinal barrier function but IgA was analyzed in serum and that effect of magnolol on serum IgA did not provide significant results. In spite this, findings may suggest the impact of magnolol in improving the gut barrier function.
Minor comments
Title
A shorter title would be advisable.
Abstract
None
Introduction
None
Results
Line 108:it is advisable to insert an "asterisk” to highlight the significant p values for all tables including Table 2
Lines 153 & 154:Table 4 refers "dorsal muscle" instead of "serum."
Line 240
-it is be advisable to provide a presumable explanation about the significant concentration of arginine in serum (but not in dorsal muscle) what would does reflect?
-did the authors run a correlation analysis? Significant correlations would strength the beneficial impact of magnolol on intestinal wellness.
-it is be advisable to state the limitations of this contribution having in mind that most biomarkers were assessed in serum not in intestinal tissue.
Material and Methods
Line 437
it is advisable to state the age of pigs, the environmental conditions of temperature, humidity, light/dark cycle, the housing conditions of piglets (barnyard or whatever). This information should be included to clarify potential concerns raised on the animal handling.
Line 444 explain on which grounds the magnolol dosage of 400 mg/kg was chosen
Line 515: see please “major comments.”
Author Response
Dear reviewer,
Thank you for your valuable comments and suggestions on our manuscript entitled “Magnolol Supplementation Alters Serum Parameters, Immune Homeostasis, Amino Acid Profiles, and Gene Expression of Amino Acid Transporters in Growing Pigs” (ijms-2567989). The comments are really valuable and super helpful for us to revise and improve our paper, as well as providing important guidance for our researches.
In light of your comments and suggestions, we have made substantial revision to our manuscript and hope that the explanation has fully addressed all of your concerns. Here, we submit the revised manuscript along with a list of our responses to the comment. To facilitate this discussion, we first retype your comments in italic font and then present our responses to the comments.
We appreciate very much for your patience and valuable comments and please let us know your further advice.
Yours sincerely,
Professor Xianyong Ma and Yulong Yin
Corresponding author
Response to Reviewer:
Reviewer #1 Comments and Suggestions for Authors…
Comment 1: A shorter title would be advisable.
Response: Thanks for your valuable advice. We have now shortened the title as follows: Magnolol Supplementation Alters Serum Parameters, Immune Homeostasis, Amino Acid Profiles, and Gene Expression of Amino Acid Transporters in Growing Pigs
Comment 2: It is advisable to insert an "asterisk” to highlight the significant p values for all tables including Table 2 (Line 108)
Response: Thanks for your valuable suggestion. We have now inserted the “asterisk” and its note in Table 2, 3 and 4 in this revision.
Comment 3: Lines 153 & 154:Table 4 refers "dorsal muscle" instead of "serum."
Response: Thanks for your careful review. We have now changed the wording “higher serum histidine” and “in serum arginine” into “higher dorsal muscle histidine” (Line 152) and “in dorsal muscle arginine” (Line 154) in this revision, respectively. To avoid the same mistake, we have checked the full manuscript.
Comment 4: It is be advisable to provide a presumable explanation about the significant concentration of arginine in serum (but not in dorsal muscle) what would does reflect? (Line 240)
Response: Thanks for your invaluable suggestion.
Arginine, a conditionally essential amino acid for young mammals’ growth, is primarily obtained from dietary sources, protein turnover and endogenous synthesis in animals[1]. Diet constitutes the primary pathway of arginine in animals, while other pathways fail to adequately fulfill their physiological requirements[2]. Therefore, we speculated that the increased serum arginine levels in this study primarily resulted from its absorption in the intestine, and substantiated our hypothesis by assessing the relative mRNA expression of arginine transporters.
The increased levels of arginine in serum can facilitate a diverse range of physiological processes. Firstly, arginine serves as a precursor for the biosynthesis of creatine, glutamic acid, and polyamines, which are essential regulators of cellular growth, proliferation and protein synthesis. Moreover, arginine could be catalyzed by nitric oxide synthase (NOS) to produce NO, which plays a regulatory role in the immune system by affecting immune cells and factors[3]. Additionally, the combined production of glutamic acid from ornithine and arginine stimulates the release of growth hormone, thereby facilitating growth. Accordingly, the enhancement of serum arginine level, to some extent, suggested the augmentation of immune function, protein synthesis, and cellular proliferation.
By measuring the amino acid profiles of the dorsal muscle, we found a significant increase in glutamate. Glutamate was mainly utilized by intestinal epithelium instead of being absorbed into the portal vein. Moreover, the body predominantly relies on de novo synthesis for the production of glutamate[4]. Previous studies have reported that arginine mainly undergoes metabolic processes via two direct pathways, namely the ornithine (Orn) pathway and citrulline pathway, and can produces glutamate and glutamine via the Orn pathway[5]. Recent studies have revealed strong activity of ornithine aminotransferase (OAT), an enzyme that catalyzes the synthesis of glutamate from ornithine and arginine, in muscle tissue[6]. Based on our results that there was a significant increase in serum arginine level, no change in serum glutamate level, no alteration in muscle arginine level, and a significant elevation in muscle glutamate level in response to magnolol supplementation, we speculated that the increased serum arginine may be utilized to synthesize glutamate in muscle via the arginine-ornithine-glutamate pathway. However, the elucidation of the mechanisms requires further investigation.
To clarify, we have now revised the presumable explanation in this revision as follows: “Glutamate, a substance centrally posited in mammalian metabolism, plays a crucial role in tissue synthesis, particularly in muscle, where it promotes intramuscular protein synthesis and facilitates rapid gain of lean tissues in growing pigs[57, 58, 59]. Reeds et al.[60] reported that dietary glutamate had no significant impact on muscle glutamate content since it was mainly utilized by intestinal epithelium instead of being absorbed into the portal vein. Moreover, the body predominantly relies on de novo synthesis for the production of glutamate. However, arginine can be catabolized to generate various compounds, such as glutamate, and glutamine[61]. Increased levels of arginine in the bloodstream may result in an increase in muscle glutamate content[62, 63]. These studies aligned with our study, where a significant increase in serum arginine levels, no change in serum glutamate levels, no altering in muscle arginine levels, and a significant elevation in muscle glutamate levels were observed in response to magnolol supplementation. Thus, the inclusion of magnolol in the pig diets led to increased levels of muscle glutamate, potentially due to its ability to increase the serum arginine concentration and subsequently promote glutamate accumulation in muscle tissue. However, further research is necessary to confirm this hypothesis.” (Line 400-416).
References cited here are as follows:
[1] Reeds, P.; Burrin, D.; Davis, T.; Fiorotto, M.; Stoll, B.; Goudoever, J., Protein Nutrition of the Neonate. P Nurt Soc 2000; 59(1), 87-97.
[2] Morris, S. M., Arginine: beyond protein. Am J Clin Nurt 2006; 83(2): 508S-512S.
[3] Stuehr DJ., Enzymes of the L-arginine to nitric oxide pathway. J Nutr 2004;134(10):2748S-2751S.
[4] Reeds, P. J.; Burrin, D. G.; Jahoor, F.; Wykes, L.; Henry, J.; Frazer, E. M., Enteral Glutamate is Almost Completely Metabolized in First Pass by the Gastrointestinal Tract of Infant Pigs. Am J Physiol 1996, 270, (3 Pt 1), E413-8.
[5] Mori, M.; Gotoh, T., Regulation of Nitric Oxide Production by Arginine Metabolic Enzymes. BBRC Medline 2000, 275(3),715-719.
[6] Ginguay, A.; Cynober, L.; Curis, E.; Nicolis, I., Ornithine Aminotransferase, An Important Glutamate-Metabolizing Enzyme at the Crossroads of Multiple Metabolic Pathways. Biology 2017, 6, 18.
Comment 5: Did the authors run a correlation analysis? Significant correlations would strength the beneficial impact of magnolol on intestinal wellness.
Response: Thanks for the suggestion. We totally agree with you. In this study, we focused on the altered amino acid profiles and the underlying mechanisms by magnolol supplementation. The majority of the indicators we examined were serum biomarkers, with limited focus on intestinal wellness parameters. Thus, we don’t run the correlation analysis due to insufficiency of the intestinal wellness-related parameters.
Comment 6: It is be advisable to state the limitations of this contribution having in mind that most biomarkers were assessed in serum not in intestinal tissue.
Response: As suggested, we have now added the limitations in this revision as follows: However, the information regarding biomarker of intestinal barrier functions in response to magnolol supplementation and the mechanisms underlying the effect of magnolol on the conversion of arginine to glutamate was limited (Line 542-544).
Comment 7: It is advisable to state the age of pigs, the environmental conditions of temperature, humidity, light/dark cycle, the housing conditions of piglets (barnyard or whatever). This information should be included to clarify potential concerns raised on the animal handling.
Response: Thanks for your valuable suggestion. We have now added the comprehensive information regarding the age and health state of pigs as well as their feeding environment and methods in this revision as follows: A total of 42 healthy 70-d-old crossbred growing barrows (Duroc × Landrace × Large White) were randomly divided into 2 groups according to their body weight (the average BW in the control and magnolol groups were 25.32 and 25.31 kg, separately). Each group consisted of 7 replicates, with 3 pigs per replicate. Pigs in 2 groups were fed the basal diet (CON) and the basal diet supplemented with 400 mg/kg magnolol (Mag), respectively. The experiment lasted for 35 d, including a 5-d adaptation period and a 30-d experimental period. The reared trial was selected from June to July at the Institute of Animal Science, Guangdong Academy of Agricultural Sciences, China. During the feeding experiment, the temperature was not controlled. The ambient temperature was within a range of 25–31 °C, and the humidity was approximately 65%, with a 13-h natural light and 11-h dark cycle. There were 14 pens in total and all pens were located in one building. Pigs resided in adjacent individual concrete pens (4.5 m × 1.6 m) with concrete floors and catered with nipple drinkers and manual metal feeders, and were allowed ad libitum access to the diet and fresh water throughout the trial. The basal diet (Supplementary Table S1) meets the nutritional requirements for NRC 2012. Health conditions and feed consumption of pigs in each replicate were recorded weekly. Magnolol (purity, ≥98%) was obtained from Hunan Zhongmao Biological Technology Co. Ltd. (Changsha, China) (Line 451-467).
Comment 8: Explain on which grounds the magnolol dosage of 400 mg/kg was chosen
Response: Thanks for the comment.
Our previous study investigated the effects of different dosage (200 and 400 mg/kg) of magnolol on weaning piglets (7.91 ± 0.00 kg) at 21 days old. The whole feeding period lasted for 42 days (7 days for adaption and 35 days for experiment). Additionally, the results showed that dietary supplementation with 400 mg/kg magnolol could significantly improve the growth performance (Table 1), lipid metabolism, and antioxidant capacity compared to both the control group and 200 mg/kg magnolol group[1, 2]. Moreover, considering the feeding cost, we consequently selected the growing pigs (70 days) to investigate the potentially beneficial effect of magnolol on amino acids turnover in pigs.
Table 1 Effect of magnolol on growth performance of weaned piglets[1]
|
Items |
Time |
Control group |
200 mg/kg magnolol |
400 mg/kg magnolol |
P-value |
|
ADFI (kg/g) |
Day 1 to 19 |
0.35 ± 0.01 |
0.32 ± 0.01 |
0.32 ± 0.01 |
0.062 |
|
|
Day 20 to 35 |
0.60 ± 0.03 |
0.62 ± 0.02 |
0.62 ± 0.03 |
0.823 |
|
ADG (kg/g) |
Day 1 to 19 |
0.31± 0.01b |
0.32 ± 0.01ab |
0.34 ± 0.01a |
0.041 |
|
|
Day 20 to 35 |
0.42 ± 0.02b |
0.45 ± 0.02ab |
0.53 ± 0.03a |
0.019 |
|
F:G |
Day 1 to 19 |
1.12± 0.02a |
1.03± 0.04ab |
0.95± 0.02b |
0.001 |
|
|
Day 20 to 35 |
1.44 ± 0.06a |
1.39± 0.08ab |
1.20± 0.05b |
0.038 |
References cited here are as follows:
[1] Qu, S.; Tian, Q.; Mei, H.; Li, Z.; Li, Y.; Ma, X.; Gao, F., Effects of Magnolol on Growth Performance, Liver Antioxidant Function and Lipid Metabolism of Weaned Piglets. Chinese j Anim Nutr.2021, 34, (05), 2872-2883.
[2] Mei, H.; Li, Y.; Tian, Q.; Li, Z.; Rong, T., Ma, X.; Tian, Z.; Cui, Y.; Yu, M. Effects of Magnolol on Nutrient Apparent Digestibility, Serum Biochemical Indices, Main Fecal Microorganism Numbers and Their Metabolite Contents of Weaned Piglets. Chinese j Anim Nutr. 2022, 34, (08), 4919-4931.
Comment 9: The major comments: The conclusions should be limited on data and include the limitations of this interesting contribution. In conclusions (line 512) authors claim that magnolol improved the intestinal barrier function but IgA was analyzed in serum and that effect of magnolol on serum IgA did not provide significant results. In spite this, findings may suggest the impact of magnolol in improving the gut barrier function.
Response: Thanks for your valuable suggestion.
IgA can be divided into three forms: monomeric IgA, polymeric IgA, and secretory IgA (sIgA), which collectively constitute two independent systems: mucosal IgA and serum IgA[1]. sIgA, as the predominant immunoglobulin in the mucosal environment, exerts inhibitory effects on microbial adhesion to the mucosal epithelium and effectively prevents intestinal microorganisms from invading systemic circulation. Upon exposure to foreign antigens, the local immune system can be stimulated to elicit an immune response resulting in the production of sIgA, independent of the central immune system. However, it is noteworthy that sIgA binding with the antigen does not induce inflammation[2]. The predominant immunoglobulins in serum were monomeric IgA (85%), while sIgA accounted for only 1%[3]. Monomeric IgA could exhibit antibody-dependent cell-mediated cytotoxicity (ADCC), antigen presentation, and the release of inflammatory mediators following mucosal wall damage[2].
Previous studies showed that arginine can stimulate the secretion of intestinal mucosal sIgA[4]. Moreover, DAO is an intracellular enzyme mainly found in the cytoplasm in the upper layer of intestinal villus cells [5]. When the mechanical intestinal barrier function is impaired, DAO is released into the bloodstream, resulting in increased DAO activity in the blood [6]. Consequently, the blood DAO usually serves as a biomarker to assess the integrity of the mechanical intestinal barrier. We speculate that the insignificant IgA results in serum may be due to the fact that magnolol supplementation can promote the integrity of the mechanical intestinal barrier in the body, mainly stimulating the secretion of intestinal sIgA, while the stimulation of serum IgA is low.
To clarify, we have now revised the discussion part of DAO in this revision as follows: DAO serves as a valuable blood biomarker to assess the integrity of the mechanical intestinal barrier, with increased levels indicating impaired mechanical barrier function. Moreover, the mechanical barrier function serves as the histological foundation that maintains intestinal barrier function and prevents the entry of harmful substances, such as bacteria and endotoxins, across the intestinal mucosa into bloodstream[38] (Line 313-317). Furthermore, we have also changed the sentence “The current study suggested that dietary supplementation with magnolol was able to enhance the absorption and utilization of AAs in growing pigs by upregulating the gene expression of key small intestinal AA transporters, improving intestinal barrier functions, and promoting anti-inflammatory ability.” into “The current study suggested that dietary supplementation with magnolol was able to enhance the absorption and utilization of AAs in growing pigs by upregulating the gene expression of key small intestinal AA transporters, promoting anti-inflammatory ability and potentially contributing to the improvement of intestinal barrier functions.” in this revision (Line 536-539).
References cited here are as follows:
[1] Steffen, U.; Koeleman, C. A.; Sokolova, M. V. et al., IgA Subclasses Have Different Effector Functions Associated with Distinct Glycosylation Profiles. Nat. Commun 2020. 11, 120.
[2] Otten, M. A.; van Egmond, M., The Fc receptor for IgA (FcαRI, CD89). Immunol. Lett. 2004, 92(1):23-31.
[3] Monteiro, R.C., The Role of IgA and IgA Fc Receptors as Anti-Inflammatory Agents. J Clin Immunol, 30
[4] Shang H. F; Wang, Y. Y.; Lai, Y. N.; Chiu, W. C.; Yeh, S. L., Effects of arginine supplementation on mucosal immunity in rats with septic peritonitis. Clin Nutr. 2004. 23(4):561-569.
[5] Meng, Y.; Zhang, Y.; Liu, M.; Huang, YK.; Zhang, J.; Yao, Q., Evaluating Intestinal Permeability by Measuring Plasma Endotoxin and Diamine Oxidase in Children with Acute Lymphoblastic Leukemia Treated with High-dose Methotrexate. Anticancer Agents Med Chem. 2016, 16(3):387-392.
[6] Wu, Q.; Liu, N.; Wu, X.; Wang, G.; Lin, L., Glutamine Alleviates Heat Stress-induced Impairment of Intestinal Morphology, Intestinal Inflammatory Response, and Barrier Integrity in Broilers. Poult Sci. 2018. 97(8):2675-2683.

Reviewer 2 Report
In this manuscript, the authors investigate the effect of supplementing growing pigs with a polyphenolic compound on performance, immune response and protein synthesis. Their results suggest that Magnolol do not improve performance but could improve pig health by increasing intestinal absorption and transport of amino acids.
This is a well-written manuscript on a relevant topic for animal production, which could contribute to the literature in the field. However, some modifications are needed, especially regarding the methods to strengthen the manuscript.
1. L215: Figure 3C: Please specify the legend for the heat map, I suppose that the colors (blue to red) correspond to the z-score?
2. L212-215: This should be in the discussion, not in the results.
3. L231-235/ Figure 4B/C: Same remark, this is an interpretation of the data and should be in the discussion. It is not very clear but I suppose that Figure 4B and C correspond to results obtained in the serum AA profile analysis? The results from the serum AA profile should not be mentioned in the metabolomics analysis part.
4. L250-260: In my opinion, this section looks more like a conclusion than the beginning of the discussion.
5. L270: Can you provide possible explanations for the conflicting results between the current study and your previous study regarding the effects of Magnolol on performance?
6. L281-284: The sentence is not clear, energy allocation towards the immune system rather than growth suggest a reduction of growth. I suppose that the authors meant that the additional energy provided by magnolol is allocated to the immune system rather than to growth but the wording is not clear, please rephrase.
7. L286-306/ L320-350: These paragraphs are 30 lines long, which make them difficult to read. It would be good to split them in separate paragraphs.
8. L371-373: It is not clear whether the HPLC results mentioned here were obtained in the current study? If so, indicate which results/figure/table does this refer to and mention in the methods that HPLC was used, if not provide reference.
9. L440: Indicate that the two groups correspond to the 2 diets.
10. L441: Several indications are missing in the methods: please indicate age, sex and body weight of the pigs in the 2 groups.
11. L442: There is no information on the pig housing: individual/collective pens, dimensions of pens, type of feeders. Were the different replicates and groups housed next to each other? It is important that no other variable may affect the animal response between groups.
12. L452: Was ADFI measured manually by weighing the refusal or automatically? If ADFI was measured per replicate, I suppose that individual ADFI was obtained by dividing the total quantity by 3. Please clarify.
13. L456: The sentence is not clear: Was the animal to be slaughtered randomly selected or was it selected in average BW?
14. L495: Please indicate which type of metabolomics was performed : LC-MS, NMR or other?
15. L512: Maybe this is a requirement from the journal (if so, the comment can be dismissed) but I find it unsettling to have the conclusion after the methods and not straight after the discussion.
Author Response
Dear reviewer,
Thank you for your valuable comments and suggestions on our manuscript entitled “Magnolol Supplementation Alters Serum Parameters, Immune Homeostasis, Amino Acid Profiles, and Gene Expression of Amino Acid Transporters in Growing Pigs” (ijms-2567989). The comments are really valuable and super helpful for us to revise and improve our paper, as well as providing important guidance for our researches.
In light of your comments and suggestions, we have made substantial revision to our manuscript and hope that the explanation has fully addressed all of your concerns. Here, we submit the revised manuscript along with a list of our responses to the comment. To facilitate this discussion, we first retype your comments in italic font and then present our responses to the comments.
We appreciate very much for your patience and valuable comments and please let us know your further advice.
Yours sincerely,
Professor Xianyong Ma and Yulong Yin
Corresponding author
Response to Reviewer:
Reviewer #2 Comments and Suggestions for Authors…
Comment 1:(L215) Figure 3C: Please specify the legend for the heat map, I suppose that the colors (blue to red) correspond to the z-score?
Response: Thanks for the suggestion. The heatmap in Figure 3C illustrated the Z-score transformed peak area of differential metabolites in the untargeted metabolome. We have now redrawn the heatmap and incorporated information of distinct color schemes to facilitate differentiation among various groups (Figure 3C). Additionally, for the sake of clarity, we have now reworded the legend of figure 3C as follows: (C) Z-score heatmap of visualizing the remarkably changed metabolites (Z score was utilized to transform the metabolites peak area) between the Con and the Mag group. Con, control group, pigs fed a basal diet; Mag, magnolol group, pigs fed a basal diet supplemented with 400 mg/kg magnolol (Line 216-219 in this revision).
Comment 2: (L212-215) This should be in the discussion, not in the results.
Response: Thanks for your valuable suggestion. We have now deleted this sentence in this revision (Line 213). To avoid the same situation, we have checked the “result” section carefully.
Comment 3: (L231-235) Figure 4B/C: Same remark, this is an interpretation of the data and should be in the discussion. It is not very clear but I suppose that Figure 4B and C correspond to results obtained in the serum AA profile analysis? The results from the serum AA profile should not be mentioned in the metabolomics analysis part.
Response: Thanks for your valuable suggestion. We have now changed Figure 4B/C in this revision as follows:
Figure 4B represented the result obtained from our UPLC-Orbitrap-MS/MS analysis, while Figure 4C illustrated the result obtained from serum AA profile (HPLC analysis).
In order to bolster the robustness of our findings that the level of arginine was significantly increased by magnolol supplementation, we employed fold change analysis obtained from the arginine concentration detection by both UPLC-Orbitrap-MS/MS (Figure 4B) and HPLC (Figure 4C).
To clarify, in this revision, we have now rephrased the description of this result in the “result” section (Line 231-238) as follows: Through analyses of key metabolites associated with AA metabolic pathway, L-arginine was identified as the most crucial metabolite influencing AA metabolic path-way. Fold change analysis obtained from the arginine concentration detection by UPLC-Orbitrap-MS/MS (Figure 4B) and HPLC (Figure 4C) was performed to bolster the robustness of the findings that the level of arginine was altered by magnolol supplementation. Magnolol supplementation significantly upregulated the levels of arginine as observed in both UPLC-Orbitrap-MS/MS and HPLC analyses (P < 0.05), resulting in respective 2.42-fold and 1.59-fold increases.
Also, we have reworded the legend of this result (Line 241-245) as follows: (B) The fold change of serum L-arginine level (the peak area) based on UPLC-Orbitrap-MS/MS analysis (Relative quantification), (C) The fold change of serum L-arginine level (the concentration, nmol/μL) based on HPLC analysis (Absolute quantification). Con, control group, pigs fed a basal diet; Mag, magnolol group, pigs fed a basal diet supplemented with 400 mg/kg magnolol.
Comment 4: (L250-260) In my opinion, this section looks more like a conclusion than the beginning of the discussion.
Response: Thanks for your invaluable and detailed suggestion. We have now revised this section as follows: Ensuring adequate nutrition during the animal’s growth period is crucial in modern agriculture to maximize genetic potential for growth[11]. Insufficient supplies of AAs during the growth phase can lead to impaired growth and decreased skeletal muscle mass[30]. Polyphenols have recently attracted enormous attention due to their ability to facilitate the absorption and transport of AAs in the small intestine[18], as well as maintain intestinal health[17]. Our preliminary results demonstrated that magnolol, a potent herbal polyphenolic compound, was able to enhance the apparent digestibility of crude protein in pigs[29]. However, the mechanisms underlying the increased protein absorption and utilization induced by magnolol remain unclear. In this study, magnolol supplementation significantly enhanced serum TP, ALB and immune factors, while also inducing alterations in both serum and muscle AA profiles. Furthermore, these alterations were associated with the upregulated expression of AA transport-related genes and the enriched metabolic pathways involved in AA synthesis and metabolism. Taken together, magnolol has the potential to serve as a nutritional supplement to enhance the growth of growing pigs by increasing protein synthesis and muscle mass while optimizing their genetic growth capacity (Line 247-262 in this revision).
Comment 5: (L270) Can you provide possible explanations for the conflicting results between the current study and your previous study regarding the effects of Magnolol on performance?
Response: Thanks for the comment. Our previous study demonstrated that dietary supplementation with 400 mg/kg magnolol considerably increased the growth performance in weaned piglets (day 21, 7.91 ± 0.00 kg), such as improving the average daily gain (ADG) (0.33 ± 0.02 vs 0.38 ± 0.02* kg/d) and reducing the feed-to-gain ratio (F/G) (1.48± 0.07 vs 1.26± 0.05*). Therefore, we hypothesize that the inclusion of 400 mg/kg magnolol in the diet may elicit a similar beneficial effect on the growth performance of growing pigs. However, there were no significant differences in the growth performance of growing pigs, including the body weight gain, average daily feed intake, final body weight, ADG, and F/G, between the control and magnolol groups. The inconsistency regarding the growth performance of pigs in response to magnolol treatment may be attributed to variations in the physiological characteristics of the experimental subjects across these two studies. Due to immature intestines and immune systems, weaning piglets are susceptible to oxidative stress, which leads to sub-health status and poor growth performance[1]. Magnolol is recognized as a potent antioxidant that can enhance the host's antioxidant capacity and maintain redox balance, thereby enhancing the growth performance of piglets by bolstering resistance against oxidative stress and ameliorating health conditions[2]. As the pig matures, its digestive, immune systems, and gut microbiota gradually attain a fully developed and stable state, thereby enhancing adaptability to both internal and external stimuli [3, 4]. Therefore, in the presence of well rearing conditions and management, it may be somewhat challenging to significantly increase the growth performance of the healthy pig by magnolol supplementation. In conclusion, the above statements provide a potential and plausible explanation for the observed variations in growth performance effects of magnolol supplementation between growing pigs and weaned piglets. However, further investigations are warranted to elucidate the precise underlying mechanisms.
References cited here are as follows:
[1] Qi M.; Tan B.; Wang J.; Liao S.; Li J.; Liu Y.; Yi Y, Post-natal Growth Retardation Associated with Impaired Gut Hormone Profiles, Immune and Antioxidant Function in Pigs. Frontiers in Endocrinology 2019;10(660.
[2] Qu, S.; Tian, Q.; Mei, H.; Li, Z.; Li, Y.; Ma, X.; Gao, F., Effects of Magnolol on Growth Performance, Liver Antioxidant Function and Lipid Metabolism of Weaned Piglets. Chinese Journal of Animal Nutrient.2021, 34, (05), 2872-2883.
[3] Li, Z.; Lu, Y.; Liu, J.; Wang, J.; Su, Y., Effects of Probiotics on the Growth Performance, Meat Quality and Colonic Microflora of Growing and Finishing Pigs. Journal of Nanjing Agricultural University.2021, 34, (05), 2872-2883.
[4] Yu, M.; Mu, C.; Zhang, C.; Yang, Y.; Su, Y.; Zhu, W., Marked Response in Microbial Community and Metabolism in the Ileum and Cecum of Suckling Piglets after Early Antibiotics Exposure. Frontiers in Microbiology 2018;9(1166.
Comment 6: (L281-284) The sentence is not clear, energy allocation towards the immune system rather than growth suggest a reduction of growth. I suppose that the authors meant that the additional energy provided by magnolol is allocated to the immune system rather than to growth but the wording is not clear, please rephrase.
Response: Thanks for your suggestion. The intended message we want to express aligns with your speculation. We have now changed the sentence “However, the current study suggested that the lack of significant impact of magnolol supplementation on the growth performance of pigs may be attributed to the robust health condition among growing pigs, or the enhancement of the immune system resulting in energy allocation towards immune responses rather than growth” into “However, our study suggested that magnolol supplementation did not significantly promote growth performance. This may be ascribed to the potential allocation of additional energy derived from feed absorption in response to magnolol supplementation towards augmenting the immune system or other pathways, rather than promoting growth.” in this revision (Line 282-286).
Comment 7: (L286-306/ L320-350) These paragraphs are 30 lines long, which make them difficult to read. It would be good to split them in separate paragraphs.
Response: Thanks for your valuable suggestion. In order to enhance the readability, in this revision, we have now split the paragraphs based on the relevance and coherence.
As for the discussion “3.2 Effects of dietary magnolol supplementation on biochemical indices of growing pigs”, we have now divided the paragraph into three sections: Ⅰ) the first section (Line 288-304 in this revision) was focused on serum protein and nitrogen metabolism; Ⅱ) the second section (Line 305-312 in this revision) was focused on lipid metabolism; Ⅲ) the last section (Line 313-324 in this revision) was focused on mechanical intestinal barrier function.
Regarding to the discussion “3.3 Effect of dietary magnolol supplementation on the immune response of growing pigs”, we have now divided the paragraph into three sections: Ⅰ) the first section (Line 326-337 in this revision) was focused on immunoglobulins G and M; Ⅱ) the second section (Line 338-348 in this revision) was focused on IL-22; Ⅲ) the last section (Line 349-362 in this revision) was focused on IFN-γ.
Comment 8: (L371-373) It is not clear whether the HPLC results mentioned here were obtained in the current study? If so, indicate which results/figure/table does this refer to and mention in the methods that HPLC was used, if not provide reference.
Response: Thanks for your invaluable suggestion. We measured the serum arginine concentration using an HPLC-L-8900 analyzer, and the concentration was present in Table 3. In order to bolster the robustness of our findings that the level of arginine was significantly increased by magnolol supplementation, we employed fold change analysis obtained from the arginine concentration detection by both UPLC-Orbitrap-MS/MS (Figure 4B) and HPLC (Figure 4C).
To clarify, we have now changed the sentence “The AA profiles of serum are shown in Table 3.” into “The AA profiles of serum detected using high performance liquid chromatography (HPLC) are shown in Table 3.” in this revision (Line 132-133). And, to further illustrate the data source of arginine, we have changed the sentence “Interestingly, high performance liquid chromatography (HPLC) analysis revealed that dietary magnolol supplementation significantly increased serum arginine levels by approximately 59%, which was also demonstrated by a corresponding significant increase in the peak area of arginine by metabolomic analysis.” into “ Interestingly, HPLC analysis (Absolute quantification, Figure 4C) revealed that dietary magnolol supplementation significantly increased serum arginine levels by approximately 59%, which was also demonstrated by a corresponding significant increased serum arginine levels obtained from UPLC-Orbitrap-MS/MS analysis (Relative quantification, Figure 4B).” in this revision (Line 379-383).
Comment 9: (L440) Indicate that the two groups correspond to the 2 diets.
Response: As suggested, we have now added the diet information in this revision as follows: Pigs in 2 groups were fed the basal diet (CON) and the basal diet supplemented with 400 mg/kg magnolol (Mag), respectively (Line 454-455).
Comment 10: (L441) Several indications are missing in the methods: please indicate age, sex and body weight of the pigs in the 2 groups.
Response: As suggested, we have now added the comprehensive information about the pigs' status in this revision as follows: A total of 42 healthy 70-d-old crossbred growing barrows (Duroc × Landrace × Large White) were randomly divided into 2 groups according to their body weight (the average BW in the control and magnolol groups were 25.32 and 25.31 kg, separately) (Line 451-453).
Comment 11: (L441) There is no information on the pig housing: individual/collective pens, dimensions of pens, type of feeders. Were the different replicates and groups housed next to each other? It is important that no other variable may affect the animal response between groups.
Response: Thanks for the suggestion. The housing conditions are as follows: There were 14 pens in total and all pens were located in one building. Pigs resided in adjacent individual concrete pens (4.5 m × 1.6 m) with concrete floors and catered with nipple drinkers and manual metal feeders. The different replicates and groups were housed as depicted in following figure.
The assigned position of Con and Mag groups in pens.
To clarify, we have now updated the description of the “4.1. Animal, diet and experimental design” in this revision as follows: “A total of 42 healthy 70-d-old crossbred growing barrows (Duroc × Landrace × Large White) were randomly divided into 2 groups according to their body weight (the average BW in the control and magnolol groups were 25.32 and 25.31 kg, separately). Each group consisted of 7 replicates, with 3 pigs per replicate. Pigs in 2 groups were fed the basal diet (CON) and the basal diet supplemented with 400 mg/kg magnolol (Mag), respectively. The experiment lasted for 35 d, including a 5-d adaptation period and a 30-d experimental period. The reared trial was selected from June to July at the Institute of Animal Science, Guangdong Academy of Agricultural Sciences, China. During the feeding experiment, the temperature was not controlled. The ambient temperature was within a range of 25–31 °C, and the humidity was approximately 65%, with a 13-h natural light and 11-h dark cycle. There were 14 pens in total and all pens were located in one building. Pigs resided in adjacent individual concrete pens (4.5 m × 1.6 m) with concrete floors and catered with nipple drinkers and manual metal feeders, and allowed ad libitum access to the diet and fresh water throughout the trial. The basal diet (Supplementary Table S1) meets the nutritional requirements for NRC 2012. Health conditions and feed consumption of pigs in each replicate were recorded weekly. Magnolol (purity, ≥98%) was obtained from Hunan Zhongmao Biological Technology Co. Ltd. (Changsha, China) (Line 451-467).
Comment 12: (L452) Was ADFI measured manually by weighing the refusal or automatically? If ADFI was measured per replicate, I suppose that individual ADFI was obtained by dividing the total quantity by 3. Please clarify.
Response: Thanks for your valuable suggestion. The feed consumption for individual replicate (pen) was weekly recorded manually. Thus, The ADFI of each pig was calculated based on the total feed intake per pen divided by the number of trial days (35 days) and the number of pigs in each pen (3 pigs).
To clarify, we have now changed the sentence “The feed intake of the pig was recorded daily for each replicate to calculate the ADFI.” into “The feed intake of pigs was recorded for each pen weekly to calculated the ADFI (per pig), which was calculated based on the total feed intake per pen divided by the number of trial days (35 days) and the number of pigs in each pen (3 pigs).” in this revision (Line 470-473).
Comment 13: (L456) The sentence is not clear: Was the animal to be slaughtered randomly selected or was it selected in average BW?
Response: Thanks for your suggestion. The animals to be slaughtered was selected in average BW. We have now changed the sentence into “At the end of the trial, a fasting period of 6 hours was implemented for all pigs, and one pig with approximately average BW (each group) was selected from each replicate.” in this revision (Line 476-477).
Comment 14: (L495) Please indicate which type of metabolomics was performed: LC-MS, NMR or other?
Response: Thanks for your scrupulous review and valuable suggestion. The experiments were performed on a Thermo Fisher Scientific UPLC system (Dionex UltiMate 3000) coupled with a mass spectrometer (Q-Exactive Focus) (LC-MS). We have now added the performed type of metabolomics as follows: The UPLC-Orbitrap-MS/MS system form Thermo Fisher Scientific (Q-ExactiveFocus, USA) was utilized to detect the serum metabolic profiles (Line 516-517).
Comment 15: (L512) Maybe this is a requirement from the journal (if so, the comment can be dismissed) but I find it unsettling to have the conclusion after the methods and not straight after the discussion.
Response: Thanks for your scrupulous review and valuable suggestion. We have reviewed the journal instructions again and make sure that this is a requirement from the journal IJMS.
